# Exploring Geometric Feature Hyper-Space in Data to Learn Representations of Abstract Concepts

**Rahul Sharma** *,† , **Bernardete Ribeiro, Alexandre Miguel Pinto † and F. Amílcar Cardoso**

Department of Informatics Engineering—University of Coimbra, 3030-290 Coimbra, Portugal;
bribeiro@dei.uc.pt (B.R.); ampinto@dei.uc.pt (A.M.P.); amilcar@dei.uc.pt (F.A.C.)
* Correspondence: rahul@dei.uc.pt
† These authors contributed equally to this work.

**Abstract:** The term concept has been a prominent part of investigations in psychology and neurobiology where, mostly, it is mathematically or theoretically represented. Concepts are also studied in the computational domain through their symbolic, distributed and hybrid representations. The majority of these approaches focused on addressing concrete concepts notion, but the view of the abstract concept is rarely explored. Moreover, most computational approaches have a predefined structure or configurations. The proposed method, Regulated Activation Network (RAN), has an evolving topology and learns representations of abstract concepts by exploiting the geometrical view of concepts, without supervision. In the article, first, a Toy-data problem was used to demonstrate the RANs modeling. Secondly, we demonstrate the liberty of concept identifier choice in RANs modeling and deep hierarchy generation using the IRIS dataset. Thirdly, data from the IoT's human activity recognition problem is used to show automatic identification of alike classes as abstract concepts. The evaluation of RAN with eight UCI benchmarks and the comparisons with five Machine Learning models establishes the RANs credibility as a classifier. The classification operation also proved the RANs hypothesis of abstract concept representation. The experiments demonstrate the RANs ability to simulate psychological processes (like concept creation and learning) and carry out effective classification irrespective of training data size.

**Keywords:** unsupervised machine learning; hierarchical learning; computational representation; computational cognitive modeling; contextual modeling; classification; IoT data modeling

## 1. Introduction

Concepts are of great value to humans because they are one of the building blocks of our recognition process. They enable us to perform cognitive functions such as classification which is fundamental in decision making and also capacitate us for contextual comprehension. The term concept has a lot to say about itself. Anything can be seen as a concept, whether it is a living being, or a thing, or an idea. An individual concept is referred to as a concrete concept (or feature) whereas a generalized form of a set of concepts (or features) can be perceived as an abstract concept. The denomination concept immediately coins the need to understand its representations. There are several conceptual representation theoretical frameworks [1] like modality-specific, localist-distributed, experience-dependent [2]. Such frameworks not only helps us to understand the various cognitive processes in humans but also the psychological ones, like creativity. Each theory has a way to represent concrete concepts through perception (or recognition), action, emotion, and introspection, but the notion of abstract concepts is debatable [1]. Abstract concepts are largely studied in psychology, and there are attempts to study them by the computational linguistics research community for Natural Language Processing (NLP) [3]. However, the representation aspect of abstract concepts is still

a challenge. In this article, we address this issue of representation of abstract concepts computationally by simulating and studying the formation of convex abstract concepts.

Computational models provide us algorithmic specificity, conceptual clarity, and precision. Besides, they empower us to perform simulations that can either be useful to test and validate psychological theories or to generate new hypotheses about how the mind works—this has turned them into an indispensable tool to study the human brain. The literature [4–6] shows that this ambitious goal is not out of reach of computational cognitive modeling. Furthermore, these types of computational tools with the ability to capture cognitive phenomena also has the potential to simulate and study some mental states and processes such as those linked to creativity [7].

Several computational modeling techniques (or tools) simulate cognitive states and represent concepts at symbolic and connectionist levels. Symbols represent information at a symbolic level. Rules are defined to manipulate symbols. Within a symbolic representation, the meaning is internal to the description itself; symbols have sense only regarding other symbols, and not regarding any real-world objects or phenomena they may represent. Adaptive Control of Thought-Rational (ACT-R) [8] is an example of symbolic approaches, with contributions in, almost, all fields of AI (such as language processing, perception, attention, decision making, etc.). At the connectionist level, information is represented by the dynamics over densely connected networks of primitive units. A particular strength of connectionist networks is their ability to adapt their behavior according to observed data. The weights among the units of a distributed network represent the learned behavior, they offer limited explanatory insights into the process, being modeled. Bioinspired Artificial Neural Networks (ANN) such as Restricted Boltzmann Machine (RBM) [9], autoencoders [10], and deep neural networks [11] are some excellent examples of connectionist approaches with a significant contribution toward classification, perception, and recognition.

A third way constitutes a hybrid view of connectionist, and symbolic methods. Connectionist Learning with Adaptive Rule Induction Online (CLARION) [12] is a methodology that is hybrid, and capable of simulating scenarios related to cognitive and social psychology. All these methodologies either require a predefined structure or have a fixed topology that imposes a limitation of having supervision, and inflexibility while modeling the concepts. Some techniques exhibit dynamic and evolving behavior while performing computational operations, such as evolving neural networks by using their genotype-phenotype mapping of cells [13]. The proposed model emulates the behavior of the dynamic creation of abstract concepts by evolving the computational model upon identifying different groups in the data.

This article proposes a computational method named Regulated Activation Network (RAN) which unifies the virtues of symbolic, distributed, and spatial representations to represent concepts (both concrete and abstract). RAN has a graph-based topology hence it is distributed, every node in the graph (network) identifies an entity, therefore it is symbolic, and every node (or entity) is viewed in an n-dimensional feature space, hence it is also spatial. The spatial view of concepts as points in multidimensional geometric feature space (see Figure 1 for six-dimensional view of concepts) is inspired by the theory of conceptual spaces [14]. The RAN's modeling has an evolving topology that enables it to build a model depicting a hierarchy of concepts. The geometrical associations among concepts aid in determining the convex abstract concepts. Further, the representatives (nodes) of the abstract concepts form a new layer dynamically, where each node acts as a convex abstract concept representative for the underlying category. Symbolically, the concepts at (relatively) lower levels in the hierarchy are identified as concrete concepts and the concepts at (relatively) higher levels are seen as abstract concepts.

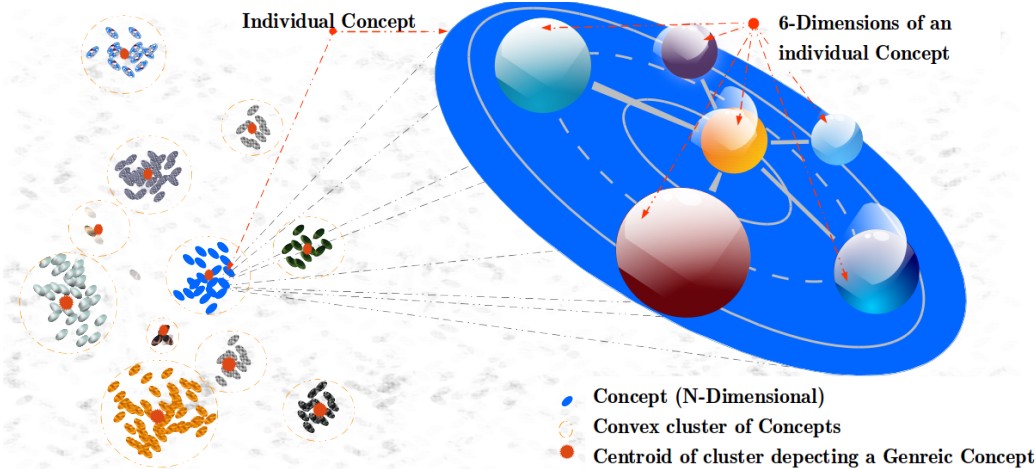

**Figure 1.** A universe of concepts in six-dimensional feature hyper-space. The ovals in the diagram depict individual concepts. Each individual concept is described by their defining six-dimensions. The cluster of concepts shows the groups formed by similar concepts represented by a convex cluster of concepts, and the cluster centers depicts the most generic concept of the cluster.

The model generation process with RAN and the three cognitive functions (i.e., concept creation, learning and activation propagation) are simulated using a Toy-data problem. The deep hierarchy generation, automatic generic concept modeling simulations are performed using two University of California Irvine (UCI) benchmarks: IRIS data; and IoT data from smartphone sensors. The application of RAN as a classifier is reported along with the proof of concept of classification using eight UCI benchmark datasets. The generated models were evaluated using metrics precision, recall, F1-score, accuracy, and Receiver Operating Characteristic (ROC) curve analysis. The article also reports the RANs classification and feature comparison with five machine learning techniques, Multilayer Perceptron (MLP) [15], Logistic Regression (LR) [16], K Nearest Neighbors (K-NN) [17], Stochastic Gradient Descent (SGD) [18] and Restrict Boltzmann Machine [9] pipelined with Logistic Regression (RBM+).

The article is organized in the following order; Section 2 puts forward the work closely related to abstract concept representation and models with evolving topology. Section 3 describes the background associated with principles, theories, and motivations for RAN modeling. RANs methodology is detailed using Toy-data in Section 4. Section 5 shows the experiments with two datasets acquired from UCI machine learning repository to exhibit (1) flexibility in choosing a suitable concept identifier, (2) building a deep hierarchy of abstract concepts, (3) automatic association of input-labels to their respective abstract concept nodes. Section 6 provides RAN comparisons with five classifiers and proof of concept with eight benchmark datasets. At last, Section 7 summarizes and concludes the article with remarks over ongoing and future work.

## 2. Related Work

Abstract concepts are of immense value because they help in developing unique abilities in humans such as relative recognition and effective decision-making. In medical science, there have been significant efforts to study abstract concepts with the help of technology. One such example is MRI (Magnetic Resonance Imaging), which is being used to inspect the sections of the brain involved in abstract concept identification [19,20]. Research in psychology has also reported investigations over abstract concepts, like probing the role of emotional content in processing and representing abstract concepts [21].

There has been a notable contribution from cognitive, and psycholinguists in studying languages through abstract concept modeling and representations. Internally representing abstract concepts

via amodal symbols like a feature list, and frames [22,23] is among the preliminary research work in linguistics. The association and context were also established, to relating abstract and Concrete words [22]. Some research reveals that we internally recognize metaphors as abstract concepts [24]. Besides theoretical methods, computational approaches are playing a vital role in comprehending and representing abstract concepts. Research in NLP addresses computational learning, comprehension and processing of human-understandable language, and its components. An interesting article published a work about the representation of abstract, and concrete concepts in daily written language using a text-based multimodal architecture of NLP [3]. Other than NLP, semantic networks are also used to study semantic similarity among abstract, and concrete nouns (of Greek, and English) [25] with the aid of network-based Distributed Semantic Model [26].

Though the aforementioned computational approaches contribute toward abstract concept modeling and representation, they have a fixed topology (i.e., the modeling process begins with a fixed structure and configuration). In connectionist computational modeling, there have been efforts to develop models that evolve. Artificial Neural Networks Adaptation: Evolutionary Learning Of Neural Optimal Running Abilities (ANNA ELEONORA) [27] demonstrated a way to grow neural networks with the aid of parallel genetic algorithms. NeuroEvolution of Augmenting Topologies (NEAT) [28] is another work that reported evolving neural network modeling, showing how nodes and weights are added to the model when new features emerge as part of the existing population and CoDeepNEAT [29] is the most recent member of such evolving models. Markov Brains [30] also belongs to the family of evolving neural networks which uses binary variables and arbitrary logic to implement deterministic or probabilistic finite state machines. They have been used to investigate behaviors, character recognition and game theory.

This article communicates an approach which is not only hybrid but also has an evolving topology. The RANs modeling learns the representation of the convex abstract concepts dynamically, hence makes it an evolving topology. RANs approach is a connectionist, and each newly created node corresponds to an abstract concept symbolically, thus portraying its hybrid characteristics.

## 3. Background

This section provides information about the principles and methodologies related to RANs modeling. It highlights the significance of each approach, along with their applicability in RANs modeling.

### 3.1. Principles of Regulated Activation Networks

The tenets of RANs modeling presented in [31], state that the model should be topologically connectionist and intend to represent and simulate the dynamic cognitive state of an agent. In the first version RAN [31] the authors implemented a single-layer version of the model where each node had a lateral connection to its same-layer companions. It had a simple learning and reasoning mechanisms, but these showed to be sufficient to simulate several known cognitive phenomena such as the Priming [32], the False Memory [33,34].

Two principles of Regulated Activation Networks inspired our proposal. First, the model should be dynamic, and this is achieved by dynamically creating layers (deep representations) of concepts. Second, the model must be capable of learning and creating an abstract representation of concepts. This is obtained by viewing associations among the concepts (at the same level) in n-dimensional geometric space, and learning relationship between the newly created abstract concepts, and input level concepts.

### 3.2. Conceptual Spaces

Conceptual Spaces Theory [14] is one of the cognitive approaches that form the basis of RANs modeling. This theory views the concepts as regions within a multi-dimensional space, with the data features representing the dimensions. The similarity among the concepts can be identified based upon

the geometrical distance between the objects. The conceptual spaces thus serve as a natural way or tool to capture the similarity relationships among concepts, or objects. Under this setting, one data instance corresponds to a single point in the space. Formally we can say, the quality dimensions, i.e., a set of $D_1, \dots, D_n$, forms the conceptual space $S$. A point in $S$ is represented by a vector $v = \langle d_1, \dots, d_n \rangle$, where $\{1, \dots n\}$ are the indexes of the dimensions. Atomic concepts are convex regions—a convex region $C$ having point $x$ that falls between points $x_1 \in C$ and $x_2 \in C$ also belongs to $C$. The quality dimension is the basic requirement for conceptual spaces [35]. An example is a color space with the dimensions Hue, Saturation, and Brightness. Each quality dimension has a geometrical structure. For example, Hue is circular, whereas brightness and saturation correspond with finite linear scales (see Figure 2).

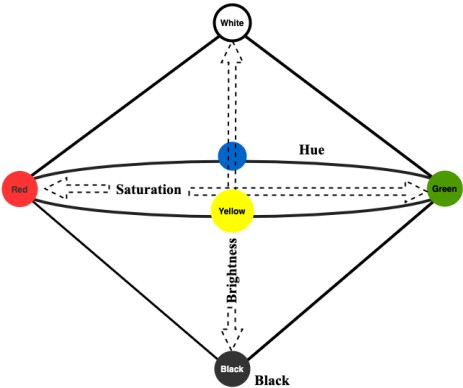

**Figure 2.** The color space [36].

The theory of conceptual spaces also addresses prototype theory of categorization [37–39]. The main idea of prototype theory is that within a category of objects, like those instantiating a concept, certain members are judged to be more representative of the group than others. For example, robins are judged to be more representative of the category "bird" than are ravens, penguins, and emus. If convex regions of conceptual space describes concepts, then prototype effect is, indeed, expected, i.e., the most likely central position of a convex region describes an abstract concept. For example, if color concepts in a convex region identified as subsets of the color space, then the central points of these regions would be the most prototypical examples of the color.

Clustering is a suitable way of identifying and learning atomic convex concepts in conceptual spaces. There are several clustering techniques, like hierarchical clustering, subspace clustering [40], partitioning relocation clustering, density-based clustering, grid-based clustering and many more. Many are frequently used in the statistical and scientific analysis of data [41,42], and in machine learning for the identification of concepts/features [43]. On the other hand, the creation of a hierarchy of sub/super-concepts is a way to represent more abstract concepts and their taxonomic-like relations. Deep learning techniques [44–48] found in the literature can also be used to create deep hierarchical representations, but usually do not interpret data as points in conceptual spaces. In the proposed approach, the clustering techniques enable us to identify categories of concepts in a conceptual space thus laying the foundation to form a layer of abstract representation of concepts.

*3.3. Spreading Activation*

Spreading Activation is a theory of memory [49] based on Collins and Quillian's computer model [50] which has been widely used for the cognitive modeling of human associative memory and in other domains such as information retrieval [51]. It intends to capture the information representation and how it is processing. According to the theory, long-term Memory is represented by nodes and associative links between them, forming a semantic network of concepts. The links characterized by a weight denotes the associative or semantic relation between the concepts. The model assumes activating one concept implies the spreading of activation to related nodes, making those memory areas more available for further cognitive processing. This activation decays over time as it spreads,

which can occur through multiple levels [52], and the further it gets the weaker it becomes. That is usually modeled using a decaying factor for activation. The method of spreading activation has been central in many cognitive models due to its tractability and resemblance of interrelated groups of neurons in the human brain [53]. This theory of Spreading Activation inspires the activation propagation mechanism in our proposal to propagate (spread) activation in the upward direction, i.e., from the input-to-abstract layer in the network. The method has its significance, i.e., in the creation of the network, and in understanding the created abstract concepts.

## 4. Abstract Concept Modeling with RANs

The proposed approach models convex abstract concepts through four core steps (i.e., Concept Identification, Concept Creation, Interlayer Learning and Upward Activation Propagation), along with one optional step (i.e., Abstract Concept Labeling). The RAN's methodology is explained using a Toy-data problem. Figure 3 shows the plot of Toy-data displaying the Cluster Representative Data Points (CRDPs) for all five classes of Toy-data (the importance of CRDP is detailed in Section 4.2). The objective of this experiment is to show how RANs build a hierarchical representation dynamically and simulate cognitive process of concept creation, learning, and activation propagation. For this experiment, it was hypothesized that the created abstract concepts symbolically represents the 5 classes of Toy-data. Classification operations were performed to prove the hypothesis which is reported at the end of this section. The notations used to describe the RAN's methodology are listed in Table 1.

**Table 1.** Notations.

| Notation | Description |
|----------|-------------|
| $W$ | Inter-Layer weight matrix |
| $A$ | Output Activation |
| $a$ | Input Activation |
| $n_a$ | Number of elements in input vector at Layer $l$ |
| $n_A$ | Number of elements in output vector at Layer $l+1$ |
| $l$ | l'th Layer representative |
| $d$ | Normalized Euclidean distance |
| $C$ | Cluster center or Centroids |
| $i, j, k$ | Variables to represent node index for input-level, abstract-level, and arbitrary node index in either of the levels, respectively |
| $t$ | Iterator variable |
| $f(x)$ | Transfer function to obtain similarity relation |

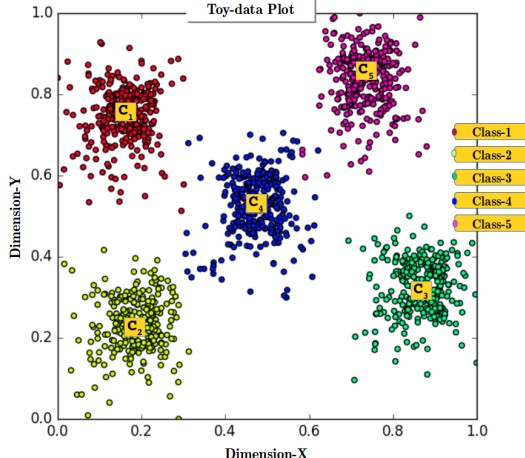

**Figure 3.** Plot of Toy-data, a 2-D artificially generated data. The plot shows five classes along with their cluster centers.

### 4.1. Assumptions and Boundaries

The necessary boundary related to input data is, data value should be between "0' and "1" (both inclusive), this limitation has its inspiration from biological neurons. A value "0" indicates neuron (or node) is inactive, whereas "1" shows the neuron is highly active. The model is, by design, applicable only to multidimensional data sets where each feature takes A real value between 0 and 1—It works as well for discrete data sets where the variables take either 0 or 1 values. If the user data is in a different format, the user must define the transformation and inverse transformation of the data. The following are a few possibilities of such alterations for some of the most common kinds of data:

- If a variable in the input data is categorical, e.g., *blue*; *green*; *red*, transform the data using One Hot Coding technique.
- If a variable in the input data is numerical, bounded within a minimum and a maximum value it can be normalized into $[0, 1]$, e.g., via $\frac{value-min}{max-min}$;

The user must implement these and the inverse transformation functions to interpret the results obtained from our model. Since our technique is designed to work with multi-variate data-sets, where each data value is a point in conceptual space, we assume that the data being used is compatible with the requirements. Though images are a form of multivariate data, pictures are not ideal candidates to be interpreted as points in conceptual spaces, (discussed in Section 3.2). For this reason, our approach will, most probably, underperform on image processing tasks against other models that are, individually, designed for these kinds of data, such as deep representations built with Convolutional Networks [47,54,55]; our technique is preferably suitable for understanding and simulating cognitive processes like abstract concept Identification. The version of RAN in this article can model data that consists of convex groups of data points, therefore, the model does not perform well well with the complex data having non-convex groups of data points. Modeling non-convex concepts is one of the ongoing research in RAN's modeling and out of the scope of this article but readers who are interested in knowing more can refer to the published research work [56].

To use the RANs approach provide the data to the model with an additional header stacked over the data. The size of the header is the same as the dimension of the input data vector, and each header element holds the largest value of their corresponding input data attribute. See Appendix A.1 for elaboration.

### 4.2. Step 1: Concept Identification (CI) Process

Concept identification is the first step in RANs modeling. The objective of the CI procedure is to appropriately identify each instance within the data as a distinguished member of various underlying convex groups. This is realized by categorizing the input data based upon their geometrical relationship, i.e., distance, conforming to the theory of conceptual spaces (see Section 3.2). Here, we also recognize data points that are the most probable representative of each identified group, complying with the prototype theory (see Section 3.2). These identified data points are termed as Cluster Representative Data Points (CRDP) and are used in Step 3 for learning the relationship between two adjacent layers (see Section 4.4).

The process of CI instantiates after preprocessing the input data. Initially, an input layer is formed, with dimension equal to the size of the input data feature vector. Step 1 in Figure 4 shows the Layer-0 with two nodes which is like the magnitude of the input vector of 2-Dimensional Toy-data. At Layer-0, clustering methods are used to determine geometrical relation among the several input data instances and identify the underlying categories within the data. Thus, K-means [57] clustering algorithm is applied to Toy-data to identify five classes (Class-1, . . . , Class-5) by assigning a value 5 to 'K' in K-means clustering algorithm (Note: The 'K' value in the K-means algorithm is to provided manually but in unlabeled datasets, the best value of 'K' can be determined using the Elbow method). Figure 3 shows the plot of 2-D data points obtained after performing concept identification operation

using the K-means algorithm. Figure 3 also displays the centroids ($C_1, \ldots, C_5$) of all the clusters, recognized as CRDPs of all five classes and will be used in Inter-Layer Learning (ILL) in Step 3 (see Section 4.4).

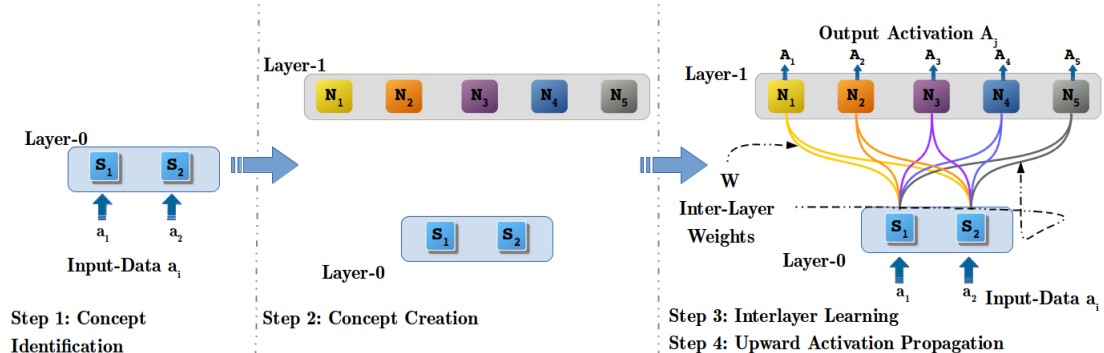

**Figure 4.** Steps in model generation with Regulated Activation Networks.

Any clustering algorithm can act as a concept Identifier in RANs modeling if it suffices two basic requirements. First, the algorithm can determine convex categories based upon their geometric relationship among the data instances. Second, the algorithm recognizes CRDPs of all the identified clusters. This flexibility of choosing a suitable method for the concept Identification process in RANs modeling is demonstrated by a separate experiment using Affinity propagation [58] clustering algorithm, in Section 5.1.

### 4.3. Step 2: Concept Creation (CC) Process

Concept creation is a cognitive process to create a representation of a newly identified concept. In RAN's modeling, this cognitive process is simulated by creating a new layer of concepts dynamically. Each constituent node in the new layer symbolically acts as an abstract representation of their respective categories identified in the CI process. The Step-2 in Figure 4 shows the newly created layer (Layer-1), that has five nodes ($N_1, \ldots, N_5$), corresponding to five classes (see Figure 3), identified in CI operation with Toy-data.

Besides abstract representation of underlying categories, the activation of nodes in newly created layer discloses the degree of confidence (DoC) (Calculating DoC of a node is explained in detail with upward activation propagation operation). indicating the certainty of identification of a class by its representative node in the new layer (for a given input data instance). For example, if a node (say $N_1$) gets an activation of 0.85, it can be stated that with a confidence of 85% the input data belongs to the category being represented by node $N_1$. Thus, for all input data instances, the obtained ⟨*feature, value*⟩ pair of ⟨*abstract-node, Activation-value*⟩ at new layer adds more meaning. For instance, in Figure 4, Step-2, at Layer-0 input vector is [0.1, 0.21] it signifies that the dimensions $S_1$ and $S_2$ has activation 0.1, and 0.21 respectively. For the, aforementioned, input vector, [0.13, 0.32, 0.89, 0.16, 0.05] vector of activation is observed at all nodes ($N_1, \ldots, N_5$) respectively, at Layer-1. The observed activation vector itself describes that the input data belongs to Class-4 with a DoC of 89%.

### 4.4. Step 3: Inter-Layer Learning (ILL) Process

Learning is an important cognitive process it acts as a relationship to associate concepts. In RANs modeling, learning is simulated by an assignment operation. The developed Inter-Layer Learning procedure also fulfills the second objective of RANs modeling (mentioned in Section 3.1). As aforestated in Section 4.3 that each node in the new layer is an abstract representative of categories identified in the CI process, thus we learn association among the two-layer such that it substantiates the abstract representation by the nodes at the new layer. Since CRDPs (see Section 4.2) are the most apparent

choice as an abstract representative of a cluster (and adhere to the inspiration from prototype theory); consequently, the CRDPs learned as an association between the two layers.

Equation (1) shows the general learning in the form of a matrix, where $W$ is the learned Inter-Layer Weight (ILW) between node $j$ at new layer (i.e., Layer-1 in Figure 4) and node $i$ at input layer (i.e., Layer-0). The set of ILWs, from one node $j$ at new layer to all input nodes $i$, are the values of CRDP of $j^{th}$ cluster center (i.e., $C_j$) identified in CI process. For instance, cluster center $C_1$ (see Figure 3) forms the weight vector [$W_{1,1}$, $W_{1,2}$, $W_{1,3}$ and $W_{1,4}$] (ILWs shown by 2 yellow lines in Step 3 Figure 4) between the node $N_1$ at Layer-1 and all four input nodes $S_1$ and $S_2$ at Layer-0.

$$
W = \begin{bmatrix} W_{1,1}, W_{1,2}, \ldots, W_{1,n_a} \\ \cdots \\ W_{k,1}, W_{k,2}, \ldots, W_{k,n_a} \\ \cdots \\ W_{n_A,1}, W_{n_A,2}, \ldots, W_{n_A,n_a} \end{bmatrix} = \begin{bmatrix} C_1 \\ \cdots \\ C_k \\ \cdots \\ C_{n_A} \end{bmatrix} \tag{1}
$$

where $j = 1, 2, \ldots, n_A$, and $i = 1, 2, \ldots, n_a$.

The distance between the learned weight vector of one node $j$ (at Layer-1) and activation of all input nodes $S_1$ and $S_2$ (at Layer-0), is used to determine how strongly the input vector represents the node $N_j$ at new layer. Thus, it enables us to identify the convex abstract concepts for the input instance (elaborated in Section 4.5).

### 4.5. Step 4: Upwards Activation Propagation (UAP) Process

This upward activation propagation is a geometric reasoning operation, i.e., a non-linear projection of an $i$-dimensional input data vector $a_i$, into a $j$-dimensional output vector $A_j$ (see Step 4 in Figure 4). The UAP operation is carried out in two stages, in the first stage the geometric distance operation takes place, and in the second stage, geometric distance is translated to establish a similarity relation.

#### 4.5.1. Geometric Distance Function (GDF)—Stage 1

In the first phase of the UAP mechanism we determine the geometrical distance between the learned weight vectors (see Equation (1)) and an input instance $a_i$. The numerator of Equation (2) shows a function to calculate the Euclidean distance between the $j^{th}$ weight vector and input vector $a_i$. The denominator of Equation (2) shows the relation that normalizes (in RANs modeling the activation values are, by definition, real values in the $[0, 1]$ interval – in an $n$-dimensional space the maximal possible euclidean distance between any two points is $\sqrt{\sum_{i=1}^{n}(a_i - 0)^2} = \sqrt{n}$, where $a_i = 1$ the calculated distance between $[0, 1]$.

$$
d_j = \frac{\sqrt{\sum_{i=1}^{n_a}(W_{j,i} - a_i)^2}}{\sqrt{n_a}} \tag{2}
$$

and consequently, $j$ normalized Euclidean distances $d_j$ are obtained between all $j$ weight vectors and input instance $a_i$.

#### 4.5.2. Similarity Translation Function (STF)—Stage 2

In the second phase the calculated normalized distance is transformed to obtain a similarity relation such that following requirements are fulfilled:

- $f(d = 0) = 1$, i.e., when distance is 0 similarity is 100%.
- $f(d = 1) = 0$ i.e., when distance is 1 similarity is 0%.
- $f(d = x)$ is continuous, monotonous, and differentiable in the $[0, 1]$ interval.

$$
f(x) = (1 - \sqrt[3]{x})^2 \tag{3}
$$

In RANs modeling Equation (3) is used as the Similarity Translation Function to determine the similarity relation of the previously calculated distance. The non-linearity of STF is depicted in Figure 5, indicating that the similarity value reduces drastically when the normalized Euclidean distance is larger than 0.05 (or 5% dissimilar).

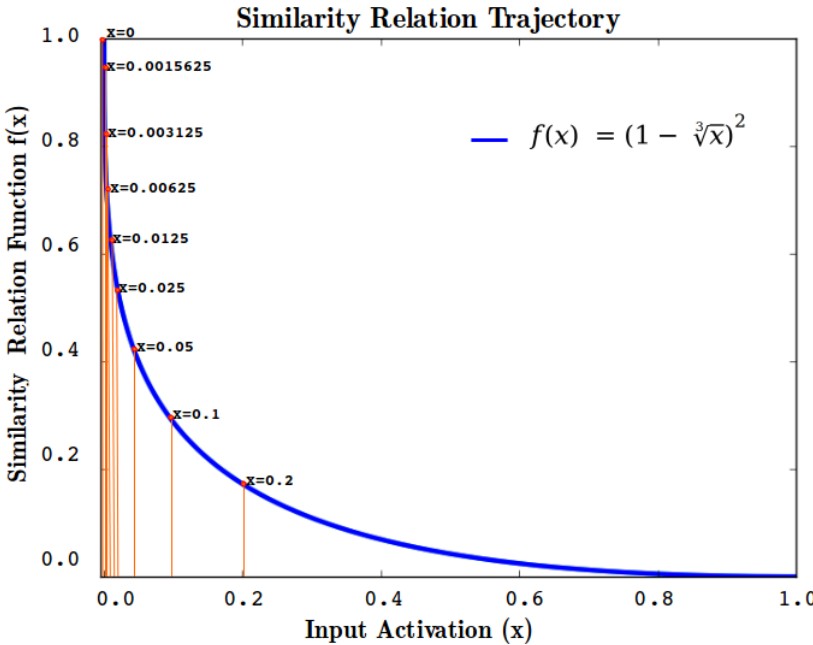

**Figure 5.** Plot of Similarity Translation Function with respect to varying input values in range $[0, 1]$.

The first three steps generate the RANs model (see Figure 4), later, in the fourth step, this model is used via UAP operation by propagating the input activation ($a_i$) upward and obtaining activation ($A_j$) at convex abstract concept layer (inspired by the theory of spreading activation see Section 3.3). Algorithm 1 describes the Upward Activation Propagation operation, showing how the inputs and interlayer learning weights $W$ are used to calculate similarity relation to generating output activation at each abstract concept representative nodes. The activation $A_j$ in newly created nodes $N_j$ also indicates the degree of confidence (DoC) of the identification of a class by its representative node in the new layer (for a given input data instance). For instance, in Figure 4, Step-2, at Layer-0 input vector is $[0.1, 0.21]$ it signifies that the dimensions $S_1$ and $S_2$ has activation 0.1 and 0.21 respectively. For the, aforementioned, input vector, a $[0.13, 0.32, 0.89, 0.22, 0.01]$ vector of activation is observed at all nodes ($N_1, \ldots, N_5$) respectively, at Layer-1. The observed activation vector itself describes that the input data belongs to Class-3 (Versicolor) with a DoC of 89%.

### 4.6. RANs Proof of Hypothesis and Complexity

At the beginning of this Section 4 it was hypothesized that nodes in the newly created layer symbolically represent abstract concepts of the five classes (Class-1, Class-2, Class-3, Class-4, and Class-5) of Toy-data. This hypothesis can be proven through classification operation using the RAN model generated with Toy-data. The classification experiment setup consists of 30 iterations of an experiment. Each experiment consist of 9 Research Design (RD)(see Table A3 in Appendix A.2), where, in every RD a 10-fold cross-validation procedure was applied. To carry out the evaluation operation *True-labels*, and *Test-labels* are determined via Abstract Concept Labeling (ACL) operation of RANs (see Appendix A.3 for ACL's description). Further, these labels were used to form a multi-class confusion matrix for the 3 classes of IRIS data and with the aid of this confusion matrix, 4 metrics (i.e., Precision, Recall, F1-Score, and Accuracy) were calculated.

---

**Algorithm 1** Upwards Activation Propagation algorithm

---

**Input:** Vector $[a_1, a_2, \dots, a_{n_a}]$ as input at layer $l$.

**Output:** New activation vector $[A_1, A_2, \dots, A_{n_A}]$ in layer $l + 1$.

**for** Each node $A_j$ in layer $l + 1$ **do**

    Calculate <u>Normalized</u> Euclidean Distance:
$$d_j = \frac{\sqrt{\sum_{i=1}^{n_a}(W_{j,i} - a_i)^2}}{\sqrt{n_a}}$$
    Transform $d_j$ through STF Equation (3):
$$A_j = f(d_j^2)$$

**end for**

*Where:*

    i = $[1, 2, \dots, n_a]$.

    j = $[1, 2, \dots, n_A]$.

    $W_{j,i}$ is ILW see Equation (1).

---

Multi-class Receiver Operating Characteristics (ROC) curves were also plotted for the five classes to support the classification experiment with Toy-data. The binary labels corresponding to the True-labels (obtained via ACL operation) were obtained using the method node-wise binary transformation of input True-label (see Appendix A.5). Further, the confidence scores for the binary vectors were calculated using the node-wise confidence-score calculation method (described in Appendix A.5).

Table 2 not only shows the RAN's comparison with the other 5 classifiers but also that RAN indeed performed well in the classification process with a performance of 99% (ca.) for all classification metrics. Figure 6 shows the ROC-AUC analysis of RANs model with Toy-data, in this graph one can see that an average AUC for all the five classes is 99% (ca.). These results show the ability of RAN's modeling to identify the abstract concept where the three nodes ($N_1$, $N_2$, $N_3$, $N_4$ and $N_5$) in Layer-1 symbolically represents the 5 classes (Class-1, Class-2, Class-3, Class-4, and Class-5) of Toy-data, respectively, as abstract concepts, hence proves the hypothesis.

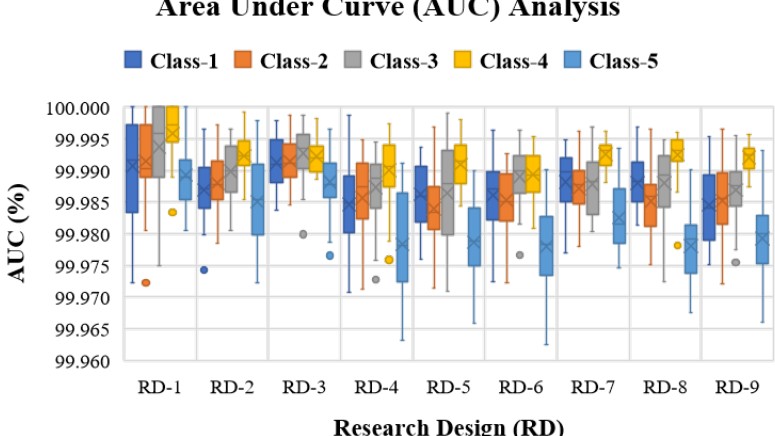

**Figure 6.** Area Under Curve for five classes of Toy-data for nine Research Designs (RD) of varying Test and Train data sizes.

**Table 2.** RAN's Comparative Study for Toy-data.

| Model | Precision (%) | Recall (%) | F1-Score (%) | Accuracy (%) |
|---|---|---|---|---|
| RBM | 90.87 ± 01.26 | 85.25 ± 2.61 | 82.34 ± 3.85 | 85.25 ± 2.61 |
| K-NN | 99.96 ± 00.08 | 99.95 ± 0.11 | 99.94 ± 0.12 | 99.95 ± 0.11 |
| LR | 99.65 ± 00.07 | 99.64 ± 0.07 | 99.64 ± 0.07 | 99.64 ± 0.07 |
| MLP | 95.62 ± 11.18 | 96.82 ± 7.56 | 96.02 ± 9.95 | 96.82 ± 7.56 |
| RANs | 99.12 ± 00.09 | 99.12 ± 0.09 | 99.12 ± 0.09 | 99.12 ± 0.09 |
| SGD | 96.00 ± 02.81 | 95.25 ± 2.86 | 94.57 ±3.76 | 95.25 ± 2.86 |

In RAN's algorithm the four operations have different complexities: (1) the concept identification process is expressed as $O(f(n))$ where $f(n)$ is the complexity of the concept identifier (or clustering algorithm); (1) the concept creation has complexity of $O(k)$ where $k$ is the number of clusters; (3) the inter layer learning also has complexity of $O(k)$ because it is an assignment operation and is equal to number of identified cluster centers; (4) the upward activation operation has the complexity of $O(n)$ when $n$ is the number data instances. The overall complexity of the RAN's modeling for creating a single layer is expressed by Equation (4).

$$T(n) = O(max \{O(f(n)), O(n)\})  \tag{4}$$

$$f(n) = O(n^{(k+2/p)})  \tag{5}$$

where: $k$ is number of clusters; p is number of features.

The time complexity of the K-means algorithm is given by Equation (5) and when K-means is chosen to be the concept identifier is, therefore, the $T(n) = O(n^{(k+2/p)})$. Table 3 lists the time complexities of all the algorithms used in this article including the RAN's time complexities with both K-means and Affinity Propagation algorithms. In Table 3 the complexities of K-means and Affinity Propagation algorithms, in fact, are the complexities of the RAN's modeling because their complexities are greater than $O(n)$.

**Table 3.** Time Complexities of Models used in the Article.

| Algorithm | Time Complexity | Description | Source |
|---|---|---|---|
| K-means | $O(n^{k+2/p})$ | $n$: n_samples; $k$: n_clusters; $p$: n_features | [59] |
| Affinity Propagation | $O(n^2)$ | $n$: n_samples | [59] |
| MLP | $O(n \cdot m \cdot h^k \cdot o \cdot i)$ | $n$: n_samples; $m$: features; $k$: no. of hidden layers; $h$: number of hidden neurons $o$: output neuron; $i$: no. of iterations | [59] |
| RBM | $O(d^2)$ | $d$: max(n_components, n_features) | [59] |
| KNN | $O(m \cdot n \cdot i)$ | $m$: n_components; $n$: n_samples; $i$: min(m, n) | [59] |
| LR | $O(n \cdot m^2)$ | $n$: n_samples; $m$: n_features | [59] |
| SGD | $O(k \cdot n \cdot \bar{p})$ | $n$: n_samples; $k$: n_iterations; $\bar{p}$: the average number of non-zero attributes per sample | [59] |

## 5. Behavioral Demonstration of RANs

This section exhibits two distinct aspects of RANs modeling via separate experiments. Both investigations present a different view of RANs methodology, highlighting the capabilities of the RANs approach.

*5.1. Experiment with IRIS Dataset*

There are two objectives of this probe, first is to demonstrate flexibility in choosing an appropriate methodology for concept Identification operation in RANs modeling (see Section 4.2). The second is to show how RANs modeling can be used to build a deep hierarchy of convex abstract concepts dynamically. This experiment uses affinity propagation [58] clustering algorithm as a concept identifier to support the claim of independence in selecting a suitable clustering method for CI process in RANs modeling. Unlike the K-means algorithm (used to describe the RANs methodology in Section 4), with the affinity propagation algorithm, the number of clusters within the data need not be known beforehand. Furthermore, affinity propagation conforms to the basic requirements (see Section 4.2) for being a concept identifier in RANs modeling.

The second prospect of this experiment is to illustrate the dynamic topology of the RAN's approach where the network grows to form several layers representing convex abstract concepts. For this demonstration, an algorithm is developed, named Concept Hierarchy Creation (CHC) algorithm (see Algorithm 2). The CHC algorithm streamlines all four steps of RANs modeling (i.e., CI, CC, ILL and UAP) and uses these steps iteratively to build a hierarchy of convex abstract concepts as described through Algorithm 2. This experiment was also conducted using the IRIS dataset obtained from the UCI machine learning repository [60]. In the CHC algorithm the affinity propagation clustering algorithm was initialized with the following parameters: (1) damping_factor (DF) = 0.94 for layers below level 3, DF = 0.9679 for the layers at level 3 and above; (2) convergence_iteration = 15; (3) max_iteration = 1000.

---

**Algorithm 2** Concept Hierarchy Creation algorithm

---

**Input:** Multi-variate data with values between [0,1].
**Output:** Set of layers of concepts—concept hierarchy.

**Initialization:** Create input layer layer-0 having dimension equal to that of input data.
Set *Current-layer-size CLS = i*, dimension of *input-data* vector.
Set *Layer-count L = 0*.
Set *Desired-depth = 6*.
Select Clustering algorithm and initialize.
Set *current-data = input-data*.
**repeat**
    Run *clustering algorithm* on *current-data* to identify set of cluster centers $C$.
    Create a *new-layer* above *current-layer*, with no nodes.
    **for** each cluster center $C_j \in C$ **do**
        *Create new node j* in *new layer l + 1*.
        **for** each node *i* in *current-layer* **do**
            Create a new weighted connection $W_{c_j,i}$
            between $c_j$ and *i* such that $W_{c_j,i}$ is the
                coordinate of *c* along the *i* dimension.
        **end for**
    **end for**
    Set *new-data* = empty data set.
    **for** each *datum* in *current-data* **do**
        Inject *datum* in *current-layer*
        Propagate activation from *current-layer* to *new-layer* using algorithm 1.
        Add activation pattern produced in *new-layer* to *new-data*.
    **end for**
    Set *L = L + 1*.
    Set *CLS* = number of clusters in *current-layer*.
    Set *current-data = new-data*.
    Set *current-layer = new-layer*.
**until** *CLS = 1* **OR** *Desired-depth = L*.

---

Input layer-0 was created, with four nodes (equal to the dimension of IRIS data), and the RANs hierarchy generation was carried out according to Algorithm 2. The model obtained from the CHC process is depicted by Figure 7, the model was initialized to grow six layers deep.

Therefore, hierarchy augmentation terminates at Layer-5, with Layer-5 identified as most abstract layer consisting of three nodes acting as abstract representatives of three categories of flowers of IRIS dataset. To evaluate the obtained RANs model, True-labels, and Test-labels were retrieved using an abstract concept labeling procedure (see Appendix A.3). A confusion matrix (see Figure 8a) was generated using the True and Test labels. With the aid of the confusion matrix, Precision, Recall, F1-Score, and Accuracy were calculated to evaluate the model. The model performed quite decently with an observed accuracy of 93.33 (ca.), the results of precision, recall, and F1-Score are reported in Table 4.

The ROC curve analysis of the RANs model, as shown in Figure 8b, displays the various operating characteristic and the observed Area Under the Curve for all the classes of IRIS data.

In this experiment, it is worth mentioning the application of RANs modeling for data dimension transformation and data visualization. In Figure 7 we can observe that the dimension of Layer-0 is four, whereas the size of the other layers either expands or reduces when the network grows. This dimension transformation operation helps address the issue of the cures of dimensionality. Besides, the transformed data can be plotted to extract useful information from the data.

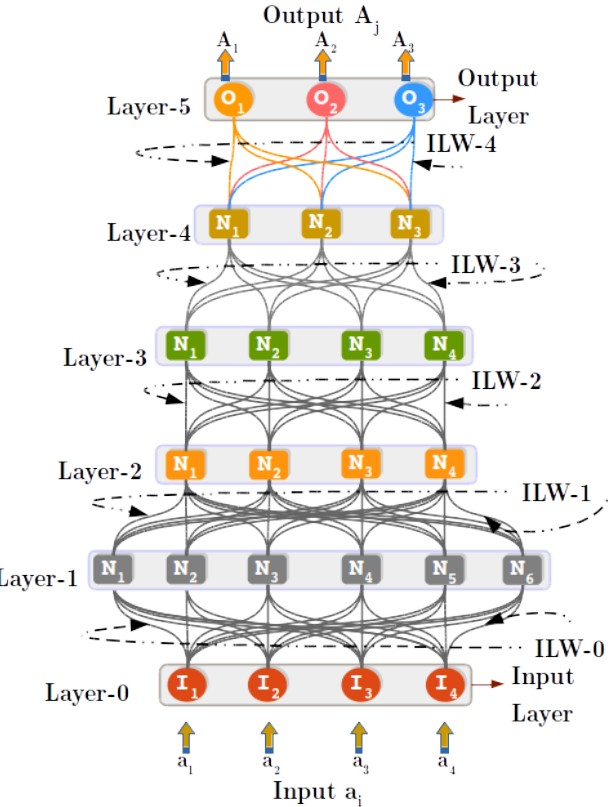

**Figure 7.** The model generated with 90% stratified IRIS data using Concept Hierarchy Creation (CHC) algorithm. Layer-0 is created while initializing the CHC algorithm. The algorithm grew to a *Desired-depth* of six Layers (including input Layer-0), and in each iteration of the CHC algorithm a new layer is created dynamically and the interlayer weights (ILW) are learned between the existing layer and a newly created layer above it.

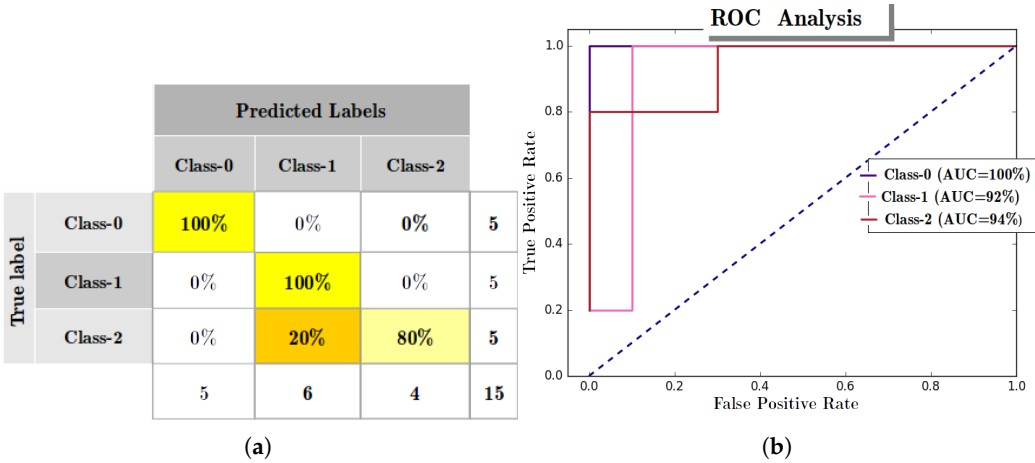

(**a**)                                                     (**b**)

**Figure 8.** Evaluation with IRIS data. (**a**) Confusion Matrix generated to validate Regulated Activated Networks (RANs) model with IRIS data (having 9 : 1 train and test data ratio) for Class-0 (Setosa), Class-1 (Verisicolour), and Class-2 (Virginica); (**b**) Receiver Operating Characteristics (ROC) curve analysis with IRIS dataset (having 9 : 1 *train, and test* data ratio), for Class-0 (Setosa), Class-1 (Verisicolour), and Class-2 (Virginica).

**Table 4.** Evaluation of RANs Model generated through IRIS data.

| Class | Precision (%) | Recall (%) | F1-Score (%) | Support |
|---|---|---|---|---|
| Setosa | 100 | 100 | 100 | 5 |
| Versicolour | 83.33 | 100 | 90.91 | 5 |
| Virginica | 100 | 80 | 88.89 | 5 |
| Avg/Total | 94.44 | 93.33 | 93.26 | 15 |

### 5.2. Experiment with Human Activity Recognition Data

This experiment aims to show the ability of the RAN's approach to building the representation of generic concepts. The experiment uses UCIHAR [61] dataset for home activity recognition using the smartphone, obtained from the UCI machine learning repository. The data captured six activities: walking, walking_upstairs, walking_downstairs, sitting, standing, and laying. The hypothesis of this experiment is that the labels walking, walking_upstairs, walking_downstairs are identified by an abstract concept (say) mobile and the other three labels sitting, standing, and laying by abstract concept (say) immobile. In this experiment also classification operation can be used to prove the hypothesis.

The UCIHAR dataset was normalized and a header was attached. In CHC algorithm K-means is chosen as a concept identifier and the parameter desired-depth was set to 1 so that model has only two layers. The K-means was configured with K = 2 because the model was hypothesized to have 2 abstract concepts at Layer-1. Having fulfilled the initialization part of the CHC algorithm modeling is performed, generating a two-layered model as depicted in Figure 9. In Figure 9 Layer-0 shows input-layer and Layer-1 corresponds to the abstract concept layer where both nodes ($N_1$, and $N_2$) represents either of the two abstract concepts (i.e., mobile and immobile abstract concepts).

Among captured six activities (walking, walking_upstairs, walking_downstairs,sitting, standing and laying), walking, walking_upstairs, and walking_downstairs are the actions of motion, whereas the remaining three represent static states. Based upon these two facts, we expect that one of the abstract nodes in Layer-1 conjointly represents walking, walking_upstairs and walking_downstairs as one class. The other node in Layer-1 stages the other three categories (i.e., sitting, standing and laying) together. Upon performing the labeling of nodes at Layer-1 through ACL procedure (see Appendix A.3 for ACL process elaboration), it was observed that walking, walking_upstairs, and walking_downstairs

classes were mapped to one node of Layer-1. Whereas, the labels sitting, standing and laying traced to the other node in Layer-1. Interestingly, this outcome commensurate with the expectations from this experiment and shows the RANs capability to identify abstract concepts in an unsupervised manner naturally.

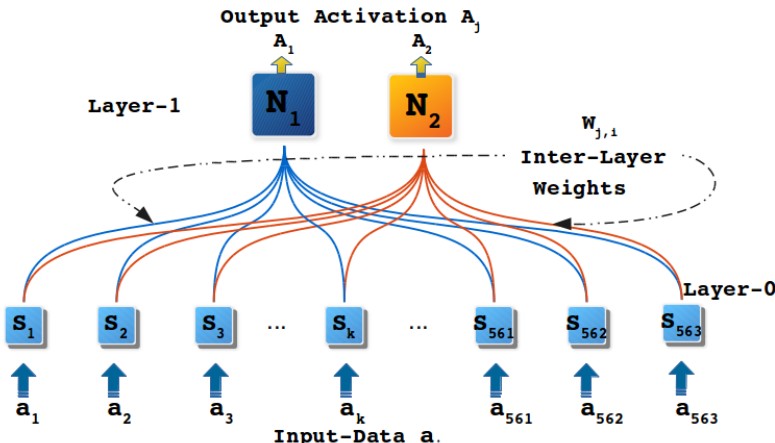

**Figure 9.** Model generated with RANs approach. Nodes $N_1$ and $N_1$ at Layer-1 represents either of the two abstract concepts, i.e., mobile and immobile. Each node at Layer-0 represents individual dimensions of input data vector.

The True-label and Test-label obtained through ACL operation were used to form the confusion matrix, which is later referred to calculate precision, recall, F1-score, and accuracy for evaluating the generated model. Node-wise binary labels and confidence scores were determined (as described in Appendix A.5) for both abstract nodes at Layer-1. Figure 10 shows the Area Under the Curve (AUC) observed during the ROC curve analysis of all 10-folds in different research designs. With both these evaluations it is deduced that, apart from building the representation of abstract concepts, the model generated with RANs performed satisfactorily.

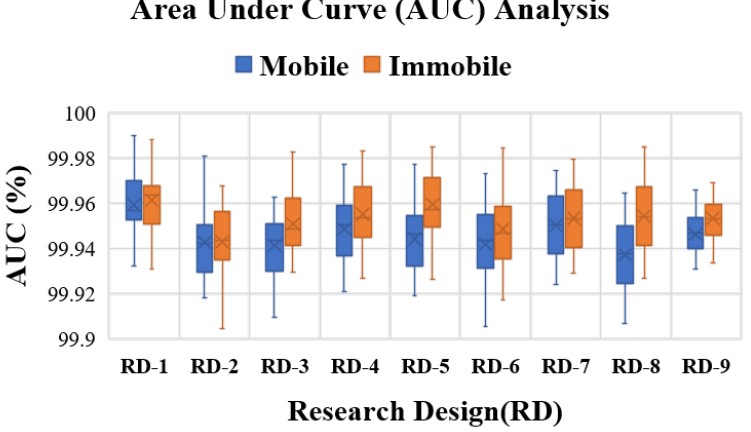

**Figure 10.** Area Under Curve observed during ROC curve analysis of UCIHAR data to determine operational points of two abstract concepts (i.e., *Mobile* and *Immobile*) for all nine Research Designs (RD).

The RANs modeling was compared with five different types of approaches based upon their classification operation. To carry out the comparative study it was essential to transform the six labels into binary labels, because RANs modeling was identifying two abstract concepts, and its performance was measured based upon them. Thus, with these five approaches, the Labels of the dataset were merged to form two groups, i.e., walking, walking_upstairs, and walking_downstairs in Class-1, and sitting, standing, and laying in Class-2. Later the modeling was performed followed by validation

and evaluation. Table 5 displays the comparison of all five approaches with RANs modeling. It is observed that RANs approach is competent to these five techniques, with an added advantage of being an unsupervised approach, and ability to build representations of abstract concepts.

**Table 5.** RAN's Comparative Study for UCIHAR dataset.

| Model | Precision (%) | Recall (%) | F1-Score (%) | Accuracy (%) |
|-------|---------------|------------|--------------|--------------|
| RBM | 99.68 ± 0.14 | 99.68 ± 0.14 | 99.68 ±0.14 | 99.68 ± 0.14 |
| K-NN | 99.96 ± 0.02 | 99.96 ± 0.02 | 99.96 ± 0.02 | 99.96 ± 0.02 |
| LR | 99.97 ± 0.02 | 99.97 ± 0.02 | 99.97 ± 0.02 | 99.97 ± 0.02 |
| MLP | 99.96 ± 0.02 | 99.96 ± 0.02 | 99.96 ± 0.02 | 99.96 ± 0.02 |
| RANs | 99.85 ± 0.01 | 99.85 ± 0.01 | 99.85 ± 0.01 | 99.85 ± 0.01 |
| SGD | 99.98 ± 0.01 | 99.98 ± 0.01 | 99.98 ± 0.01 | 99.98 ± 0.01 |

## 6. RANs Applicability and Observations

This section highlights the scope of RANs modeling as a classifier with respect to distinct domains. To support this ambit of RANs usability, experimental results are reported using eight datasets concerning different areas. A comparative study was also carried out using these datasets to match RANs classification ability with five different classifiers. Table A5 in Appendix A.5 shows configurations of all the models for all the experiments. Table A4 in Appendix A.4 provide the details about the all the datasets used in this article.

Among the eight datasets (Appendix A.4 lists the description of all the datasets used in the article), the Mice Protein [62], *Mammographic Mass* [63], Breast Cancer 569 and 669 [64,65] data pertain to the medical field, Glass Identification [66] data representing forensic science, Credit Approval [67] represents economic data, Iris [68] is a botanical data set, and Wine Recognition [69] is a data set for chemical composition analysis. The experiments performed with these datasets were akin to the investigations done with Toy-data (in Section 4), and UCIHAR data (in Section 5.2), i.e., K-means algorithm used as concept identifier, where 'K' is the number of class labels of each dataset, the hierarchy is set to have a depth of two layers (one Input and one abstract concept layer). For every dataset, models were generated using thirty iterations in nine Research Designs (RDs) (refer the Table A3 in Appendix A.2). In every RD 10-Fold cross-validation was applied to determine the performance of the models. An aggregate of precision, recall, F1-Score, and accuracy of all folds in all RDs was calculated for all the datasets, as shown in Figure 11a. From the Figure 11a it can be observed that with *Mice Protein* data RANs scores 99.99% (ca.) for all evaluation metric, whereas for *Iris, Glass Identification, Breast Cancer, and Wine Recognitions* the observations were convincing, i.e., above 89.00% (ca.). In all the folds of nine RDs ROC curves were also plotted for each class label of the eight datasets, the mean AUC for each class of the datasets is shown in Figure 11b. The evaluation metrics and ROC-AUC analysis (Figure 11a,b respectively) display the RAN's capability in machine learning tasks with different kind of datasets.

The same procedure was applied to obtain average Precision, Recall, F1-Score and Accuracy for all the datasets with five other classifiers (i.e., *RBM+, KNN, LR, MLP, and SGD*). Table 6 shows the overall comparison. It is worth noting that being dynamic and unsupervised RANs modeling performed quite satisfactorily especially with Mice Protein data, where it outperformed SGD and RBM+, was found competent with LR, KNN and MLP classifiers.

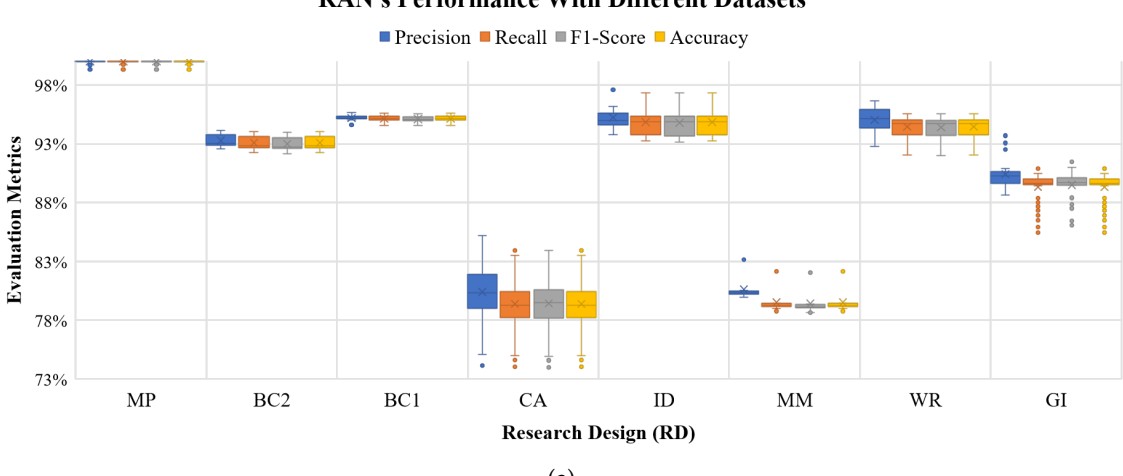

(**a**)

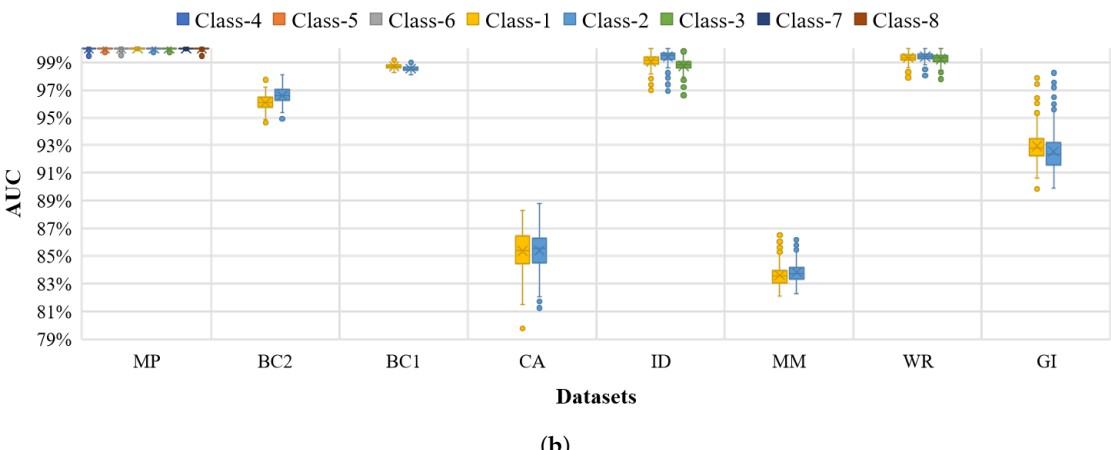

(**b**)

**Figure 11.** RANs performance with eight datasets using Precision, Recall, F1-Score and Accuracy along with ROC-AUC analysis with Eight benchmark datasets [Mice Protein (MP), Breast Cancer 669 (BC1), Breast Cancer 569 (BC2), Credit Approval (CA), IRIS data (ID), Mamographic Mass (MM), Wine Recognition (WR) and Glass Identification (GI)]. (**b**) shows the plot of percentage AUC for classes 1 to 8. For each dataset class labels of the graph is serially mapped as: *Mice protein (c-CS-s [Class-1], c-CS-m [Class-2], c-SC-s [Class-3], c-SC-m [Class-4], t-CS-s [Class-5], t-CS-m [Class-6], t-SC-s [Class-7] and t-SC-m [Class-8]); Mammographic Mass (Benign [Class-1] and Malignant [Class-2]); Credit Approval (Postitive [Class-1] and Negative [Class-2]); IRIS) (Setosa [Class-1], Versicolar [Class-2] and Verginica [Class-3]); Breast Cancer 569 (Benign [Class-1] and Malignant [Class-2]); Breast Cancer 669 (Benign [Class-1] and Malignant [Class-2]), Wine Recognition (Class-1, Class-2 and Class-3) Glass Identification (Window Glass [Class-1] and Non-Window Glass [Class-2]).* (**a**) RANs performance with eight different datasets depicting RANs appositeness with data belonging to distinct domains; (**b**) Observed Area Under Curve (AUC) while performing ROC curve analysis for RANs model generated with eight different datasets.

Figure 12 shows four graphs depicting RANs performance with different benchmark data sets. These graphs display an important aspect of RANs modeling and its performance behavior when evaluated to different research design Figure 12. The precision, recall, F1-Score, and accuracy trajectories of Human Activity Recognition (HAR), Breast Cancer 669 (BC1), Toy-data (TD) and Mice Protein (MP) Data is almost straight. The evaluation plots of Glass Identification (GI), Wine Recognition (WR), Mammographic Mass (MM), Breast cancer 569 (BC2) and Mice Protein (MP) datasets show a minimal decline in observations w.r.t RD-1 and RD-9 Research Design. On the contrary, results

from IRIS Data (ID) and Credit Approval (CA) dataset depicted a higher value while comparing the evaluation of RD-1 with RD-9 Research Designs of these data sets. Principally, the results of all four metrics of evaluation obtained similar results (with marginal variation) irrespective of the Test and Train data ratio. This is a notable observation because it shows that the RAN's approach obtains a satisfactory result even when trained with a small amount of data.

**Table 6.** RANs comparison with eight datasets belonging to different domains.

| Data | Algo | Precision (%) | Recall (%) | F1-Score (%) | Accuracy (%) | Data | Algo | Precision (%) | Recall (%) | F1-Score (%) | Accuracy (%) |
|---|---|---|---|---|---|---|---|---|---|---|---|
| Mice Protein | RBM+ | 43.45 ±44.07 | 53.50 ± 38.23 | 45.46 ± 43.36 | 53.50 ± 38.23 | Breast Cancer 569 | RBM+ | 93.60 ± 2.69 | 93.51 ± 2.77 | 93.46 ± 2.86 | 93.51 ± 2.77 |
| | KNN | 98.63 ± 3.97 | 98.34 ± 4.84 | 98.07 ± 5.65 | 98.34 ± 4.84 | | KNN | 99.80 ± 0.59 | 99.79 ± 0.62 | 99.78 ± 0.63 | 99.79 ± 0.62 |
| | LR | 98.99 ± 1.94 | 98.28 ± 3.38 | 98.14 ± 3.71 | 98.28 ± 3.38 | | LR | 99.89 ± 0.07 | 99.89 ± 0.07 | 99.89 ± 0.07 | 99.89 ± 0.07 |
| | MLP | 98.54 ± 2.19 | 98.23 ± 2.71 | 97.83 ± 3.34 | 98.23 ± 2.71 | | MLP | 98.67 ± 0.94 | 98.65 ± 0.96 | 98.64 ± 0.96 | 99.89 ± 0.07 |
| | RAN | 99.98 ± 0.06 | 99.97 ± 0.06 | 99.89 ± 0.06 | 99.97 ± 0.06 | | RAN | 93.17 ± 0.36 | 92.97 ± 0.36 | 92.87 ± 0.42 | 92.97 ± 0.36 |
| | SGD | 99.11 ± 1.84 | 98.84 ± 2.46 | 98.68 ± 2.81 | 98.84 ± 2.46 | | SGD | 99.87 ± 0.13 | 99.85 ± 0.18 | 99.83 ± 0.20 | 99.85 ± 0.18 |
| Breast Cancer 669 | RBM+ | 95.72 ± 3.62 | 95.34 ± 4.60 | 95.13 ± 5.16 | 95.34 ± 4.60 | Credit Approval | RBM+ | 76.44 ±12.50 | 75.63 ±12.98 | 74.04 ±14.59 | 75.63 ±12.98 |
| | KNN | 99.46 ± 0.88 | 99.44 ± 0.93 | 99.43 ± 0.94 | 99.44 ± 0.93 | | KNN | 95.48 ± 0.16 | 95.46 ± 0.17 | 95.46 ± 0.17 | 95.46 ± 0.17 |
| | LR | 99.16 ± 0.17 | 99.14 ± 0.17 | 99.15 ± 0.17 | 99.14 ± 0.17 | | LR | 95.06 ± 0.38 | 95.04 ± 0.39 | 95.04 ± 0.39 | 95.04 ± 0.39 |
| | MLP | 98.96 ± 0.76 | 98.95 ± 0.76 | 98.95 ± 0.77 | 98.95 ± 0.76 | | MLP | 98.02 ± 1.32 | 98.00 ± 1.34 | 97.99 ± 1.34 | 98.00 ± 1.34 |
| | RAN | 95.18 ± 0.25 | 95.15 ± 0.24 | 95.11 ± 0.25 | 95.15 ± 0.24 | | RAN | 80.67 ± 1.37 | 79.58 ± 1.05 | 79.66 ± 1.13 | 79.58 ± 1.05 |
| | SGD | 99.88 ± 0.16 | 99.88 ± 0.16 | 99.18 ± 0.16 | 99.88 ± 0.16 | | SGD | 99.77 ± 0.39 | 99.75 ± 0.40 | 99.75 ± 0.40 | 99.75 ± 0.40 |
| Glass Identification | RBM+ | 82.58 ±10.29 | 84.19 ± 4.90 | 80.61 ± 8.42 | 84.19 ± 4.90 | Mamographic Mass | RBM+ | 84.85 ±16.54 | 85.18 ±14.98 | 82.42 ±20.30 | 85.18 ±14.98 |
| | KNN | 94.08 ±12.12 | 95.97 ± 7.32 | 94.82 ±10.59 | 95.97 ± 7.32 | | KNN | 99.65 ± 0.88 | 99.64 ± 0.89 | 99.64 ± 0.89 | 99.64 ± 0.89 |
| | LR | 99.52 ± 0.18 | 99.49 ± 0.18 | 99.49 ± 0.18 | 99.49 ± 0.18 | | LR | 99.41 ± 0.30 | 99.40 ± 0.30 | 99.40 ± 0.30 | 99.40 ± 0.30 |
| | MLP | 93.78 ± 1.40 | 93.28 ± 1.52 | 92.85 ± 1.64 | 93.28 ± 1.52 | | MLP | 98.91 ± 2.11 | 98.79 ± 2.35 | 98.79 ± 2.35 | 98.79 ± 2.35 |
| | RAN | 90.07 ± 0.43 | 89.18 ± 1.23 | 89.32 ± 1.10 | 89.18 ± 1.23 | | RAN | 80.28 ± 0.18 | 79.20 ± 0.23 | 79.08 ± 0.24 | 79.20 ± 0.23 |
| | SGD | 97.95 ± 0.66 | 97.87 ± 0.69 | 97.82 ± 0.70 | 97.87 ± 0.69 | | SGD | 99.96 ± 0.03 | 99.94 ± 0.07 | 99.93 ± 0.09 | 99.94 ± 0.07 |
| IRIS | RBM+ | 79.81 ±11.91 | 77.41 ±11.88 | 70.66 ±16.28 | 77.41 ±11.88 | Wine Recognition | RBM+ | 56.00 ±25.66 | 67.05 ±16.91 | 59.07 ±21.91 | 67.05 ±16.91 |
| | KNN | 90.41 ±28.77 | 92.80 ±21.61 | 91.00 ±27.01 | 92.80 ±21.61 | | KNN | 90.74 ±26.00 | 92.88 ±19.48 | 91.14 ±24.70 | 92.88 ±19.48 |
| | LR | 97.38 ± 4.15 | 96.64 ± 5.65 | 96.45 ± 6.12 | 96.64 ± 5.65 | | LR | 94.14 ± 1.55 | 93.13 ± 1.82 | 93.00 ± 1.92 | 93.13 ± 1.82 |
| | MLP | 97.31 ± 0.71 | 96.86 ± 1.13 | 96.81 ± 1.21 | 96.86 ± 1.13 | | MLP | 97.44 ± 0.51 | 97.33 ± 0.59 | 97.32 ± 0.59 | 97.33 ± 0.59 |
| | RAN | 95.43 ± 0.67 | 95.02 ± 0.94 | 94.98 ± 0.98 | 95.02 ± 0.94 | | RAN | 94.87 ± 0.91 | 94.34 ± 1.00 | 94.29 ± 1.01 | 94.34 ± 1.00 |
| | SGD | 94.47 ± 6.40 | 94.46 ± 5.20 | 93.31 ± 6.78 | 94.46 ± 5.20 | | SGD | 98.13 ± 0.70 | 97.91 ± 0.75 | 97.91 ± 0.76 | 97.91 ± 0.75 |

Besides classification comparison, the RAN's modeling is compared with the five classifiers based upon seven features: (1) Whether the modeling in graph-based; (2) whether the modeling has a dynamic topology; (3) and (4) whether modeling can reduce or expand the dimension of the data; (5) whether modeling can perform classification; and (7) whether modeling is biologically inspired or not. Table 7 details this comparative study. It can be observed from this table that RAN is closely related to the models that are biologically inspired i.e., RBM and MLP.

**Table 7.** Feature based comparative study of RANs with five modeling techniques.

| Features\Models | RBM | K-NN | LR | MLP | RANs | SGD |
|---|---|---|---|---|---|---|
| Graph-Based | Yes | No | No | Yes | Yes | No |
| Dynamic Topology | No | No | No | No | Yes | No |
| Dimension Reduction | Yes | Yes | No | Yes | Yes | No |
| Dimension Expansion | May be | No | No | May be | Yes | No |
| Unisupervised | Yes | No | No | No | Yes | No |
| Supports Classification | Yes | Yes | Yes | Yes | Yes | Yes |
| Bio-inspired | Yes | No | No | Yes | Yes | No |

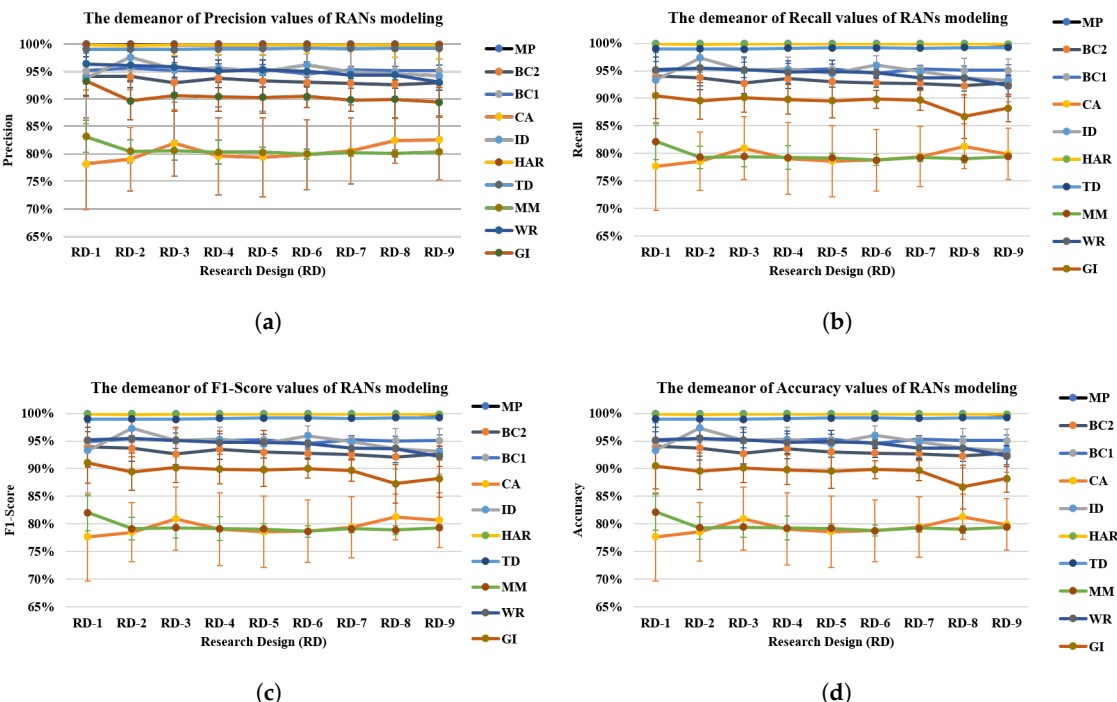

**Figure 12.** RANs evaluation metric (precision, recall, F1-Score and accuracy) value behavior with respect to varying test and train data ratio over ten datasets [Mice Protein (MP), Breast Cancer 669 (BC1), Breast Cancer 569 (BC2), Credit Approval (CA), IRIS data (ID), Mamographic Mass (MM), Human Activity Recognition (HAR), Toy-data(TD), Wine Recognition (WR) and Glass Identification (GI)] (**a**) Precision; (**b**) Recall; (**c**) F1-Score; (**d**) Accuracy.

## 7. Conclusions and Future Work

To comprehend and reasoning for emotions, ideas, etc., it is evident to understand abstract concepts because they are perceived differently from concrete concepts. There have been notable efforts to study Concrete concepts (features like walking or ingredients), but progress in investigating abstract concepts (generic features such as is-moving or recipe) is relatively less. This article proposes an unsupervised computational modeling approach, named Regulated Activation Networks (RANs), that has an evolving topology and learns a representation of abstract concepts. The RAN's methodology was exemplified through a UCI's IRIS dataset, yielding a satisfactory performance evaluation of 95% (ca.) for precision, recall, F1-Score and accuracy metrics, along with an average AUC of 99% (ca.) for all the three classes in the dataset. These evaluation result not only showed the classification capability of RANs but also proved the hypothesis of the experiment i.e., the three newly created nodes in the Layer-1 symbolically represent the three classes of IRIS data as abstract concepts.

Another experiment with IRIS data displayed the characteristic of RAN's deep hierarchy generation and independence in choosing the concept Identifier. With the aid of the Concept Hierarchy Creation algorithm (proposed in Section 5.1), the evolving nature of RAN's modeling is shown using the Affinity Propagation clustering algorithm (as an alternate concept Identifier instead of the K-means algorithm as used in modeling with a Toy-data problem). With the generated model it was shown that the model dynamically grew to a depth of six layers and performed with Precision of 94.44% (ca.), Recall of 93.33% (ca.), F1-Score of 93.26% (ca.) and Accuracy of 93.33% (ca.), along with an observed AUC of 100% (ca.), 92% (ca.) and 94% (ca.) for the three classes of data. This experiment also highlights the application of RANs modeling in data dimension transformation and data visualization.

Modeling with UCI's IoT based Home Activity Recognition (UCIHAR) smartphone sensor dataset exhibited the RAN's behavior of natural identification of generic concepts. The experiment hypothesize that six data labels (activity of walking, walking_upstairs, walking_downstairs,

sitting, standing and laying) of the dataset are to be identified as mobile (walking, walking_upstairs and walking_downstairs) and immobile (sitting, standing and laying) abstract concepts. This hypothesis was also proven using classification operation, where, the evaluation of the model shown a performance of 99.85% (ca.) for all four metrics and AUC of 99.9% (ca.) for both abstract concepts. The experiment also demonstrates how RAN can be used to model the data from the IoT domain in an unsupervised manner.

The proof of concept of RAN's modeling as a Machine Learning classifier was also provided with eight UCI benchmarks. It was identified that RAN's approach performed satisfactorily displaying the best outcome of 98.9% (ca.) with Mice Protein dataset (for all metrics). The comparison of RAN's modeling with five classifiers substantiated the effectiveness of the proposed methodology. We also observed that the RAN's performance remained similar irrespective of the size of train data. RAN was also compared with the five classifiers based upon its features and it was observed that RAN was similar to bio-inspired models. The model presented in this article is capable of modeling data that is convex which limits the RAN's performance with non-convex (or complex) datasets. As future work, we intend to improve RANs modeling that can capture the non-convexity in the data and enhance the performance of the model with complex datasets.

**Author Contributions:** R.S. performed state of the art, developed and implemented the methodology, carried out data selection and methodology validation, and prepared the original draft of the article. B.R. supervised the research work performed the formal analysis, review and editing, took care of funding. A.M.P. conceived the study plan and methodology, supervised the investigation, methodology development and implementation. F.A.C. supervised the research work, performed formal analysis, review and edition, managed funding. All authors have read and agreed to the published version of the manuscript.

**Funding:** The work presented in this paper was partially carried out in the scope of the SOCIALITE Project (PTDC/EEI-SCR/2072/2014), co-financed by COMPETE 2020, Portugal 2020—Operational Program for Competitiveness and Internationalization (POCI), European Union's ERDF (European Regional Development Fund), and the Portuguese Foundation for Science and Technology (FCT).

**Conflicts of Interest:** The authors declare no conflict of interest.

## Abbreviations

The abbreviations used in this manuscript:

| | |
|---|---|
| ACL | Abstract Concept Labeling |
| AUC | Area Under Curve |
| BC1 | Breast Cancer 669 Dataset |
| BC2 | Breast Cancer 569 Dataset |
| CA | Credit Approval Dataset |
| CHC | Concept Hierarchy Creation |
| CI | Concept Identification |
| CLS | Current Layer Size |
| CRDP | Cluster Representative Data Point |
| DoC | Degree of Confidence |
| GDF | Geometric Distance Function |
| GI | Glass Identification Dataset |
| HAR | Human Activity Recognition Data |
| ID | IRIS Dataset |
| ILL | Inter Layer Learning |
| ILW | Inter Layer Weights |
| K-NN | K Nearest Neighbor |

| MLP | Multilayer Perceptron |
|-----|------------------------|
| MM | Mammography Mass Dataset |
| MP | Mice Protein Dataset |
| MRI | Magnetic Resonance Imaging |
| RANs | Regulated Activation Networks |
| RBM | Restricted Boltzmann Machine |
| RBM+ | RBM pipe-lined with Logistic Regression |
| ROC | Receiver Operating Characteristic |
| SGD | Stochastic Gradient Descent |
| STF | Similarity Translation Function |
| UAP | Upward Activation Propagation |

## Appendix A

### Appendix A.1. Data and Scripts

This section provides links to download the data and python script used to perform RANs modeling experiments, mentioned in this article. The data and script folders can be downloaded from the web URL mentioned in Table A1. The data folder contains many files and the direct path to the files are provided in the Table A1. Similarly, the script folder *RAN_V2.0* also contains many folders where Folder *RAN* consist of the python scripts. The folder *Observations* is for storing the outcome of the experiments, at the beginning of each experiment the empty folder in directory *empty_passes_for_Experiment_Observations* must be copied into the *Observation* directory. The python script related to RANs modeling is in folder *RAN*, the description is mentioned in the Table A1.

**Table A1.** Data and Python Script of RANs modeling.

| Type | Description | File-path |
|------|-------------|-----------|
| Data | Download link | https://www.dropbox.com/sh/3410ozeru3o5opm/AAA24aUGtUS1i7xHKp9kyzRKa?dl=0 |
| | IRIS Data | data/iris_with_label.csv |
| | Mice Protein data | data/Data_cortex_Nuclear/mice_with_class_label.csv |
| | Glass Identification data | data/newDataToExplore/new/GlassIdentificationDatabase/RANsform.csv |
| | Wine Recognition data | data/newDataToExplore/new/WineRecognitionData/RansForm.csv |
| | Breast cancer 669 data | data/newDataToExplore/new/breastCancerDatabases/699RansForm.csv |
| | Breast Cancer 559 data | data/newDataToExplore/new/breastCancerDatabases/569RansForm.csv |
| | UCIHAR data | data/UCI_HAR_Dataset.csv |
| | Mamographic Mass data | data/newDataToExplore/new/MammographicMassData/RansForm1 |
| | Credit Approval data | data/newDataToExplore/new/CreditApproval/RansForm.csv |
| | Toy-data data | data/toydata5clustersRAN.csv |
| Script | Download Link | https://www.dropbox.com/sh/rcw1cj4ce1f3zic/AAAm6wVTj2qsLZ1lbc3kn4MPa?dl=0 |
| | RANs classes and methods | RAN_V2-0/RAN/RAN_kfold.py |
| | Methods | RAN_V2-0/RAN/Layer.py |
| | Utilities like Labeling and plotting | RAN_V2-0/RAN/UtilsRAN.py |
| | Python Script for using RANs | RAN_V2-0/RAN/RAN_input_T1.py |

The implemented RANs modeling tool in python takes input data in a specific format (shown in Table A2). Besides the data, the inputs require a header as the first row stacked over the original data. Each header element, $[H-1, H-2, ......., H-n]$, is the Maximum value possible for their respective column (feature, or dimension). It is assumed that the minimum value of the column is zero, if it is not then the data must be transformed between zero and the maximum positive value as described in Section 4.1.

**Table A2.** Input Data Format for implemented RANs Modeling.

| Header | H-1 | H-2 | .............. | H-n |
|---|---|---|---|---|
| | D-1 | D-2 | .............. | D-n |
| | D-1 | D-2 | .............. | D-n |
| **Data Instances** | . | . | .............. | . |
| | . | . | .............. | . |
| | . | . | .............. | . |
| | D-1 | D-2 | .............. | D-n |

*Appendix A.2. Model Configurations and Research Design*

Various experiments, reported in this article, were conducted with several datasets, using six modeling techniques including the proposed methodology i.e., RANs modeling. Table A5 in Appendix A.5 shows configurations of all the models for all the experiments. The experiments were carried out using python programing language, and implementations of Restricted Boltzmann Machine pipelined with Logistic Regression (RBM+), Logistic Regression (LR), K-Nearest Neighbor (K-NN), Multilayer Perceptron (MLP), and Stochastic Gradient Descent (SGD) models of Scikit-learn library [59]. It is to be noted that experiments with RBM were carried out, pipelined with the LR algorithm using the default configuration of its implementation in scikit-learn library. The Table A3 lists the nine Research Designs (RD) used in the experiments of this article. In every RD the ratio of the Train and Test data is varied to capture the ability of the classifier being inspected.

**Table A3.** Train and Test data distributions in nine Research Designs (RDs).

| RD-1 | | RD-2 | | RD-3 | | RD-4 | | RD-5 | |
|---|---|---|---|---|---|---|---|---|---|
| *Train* | *Test* | *Train* | *Test* | *Train* | *Test* | *Train* | *Test* | *Train* | *Test* |
| 90% | 10% | 80% | 20% | 70% | 30% | 60% | 40% | 50% | 50% |

| RD-1 | | RD-7 | | RD-8 | | RD-9 | |
|---|---|---|---|---|---|---|---|
| *Train* | *Test* | *Train* | *Test* | *Train* | *Test* | *Train* | *Test* |
| 40% | 60% | 30% | 70% | 20% | 80% | 10% | 90% |

*Appendix A.3. Abstract Concept Labeling (ACL)*

This method is optional and useful when the input data is labeled. With this mechanism, we associate an identifier to every Abstract concept node $N_j$. Having generated the RANs model with CI, then trough CC, ILL, input data is sorted label-wise, and perform UAP operation. The propagated data is inspected class-wise, and label node $N_j$ with a class-name for which it got the maximum count of the highest activation. For example, suppose input data for class-*X* has 100 instances, after inspecting the propagated data, it is observed that node $N_1$ received highest activation 74-times, whereas, with remaining 26 cases other nodes experienced maximum activation, therefore, we recognize node $N_1$ as representative of class-*X*. *True-Labels* are identified by mapping each class of the input instance directly to its respective node representative *Observed-Labels* are obtained by propagating every test-instance through UAP operation, inspecting which Abstract node received the highest activation for that data-unit, and label it with the class represented by that node. True-Labels and Observed-Labels are used to validate the model's performance.

*Appendix A.4. Dataset Description*

**Table A4.** Dataset description.

| Dataset | | | | Attribute | | Class | Source |
|---|---|---|---|---|---|---|---|
| Name | Type | Size | Balanced | Type | Size | # | Name |
| Mice Protein | Multivariate | 1080 | yes | Real | 82 | 8 | UCI |
| Breast Cancer 569 | Multivariate | 569 | yes | Real | 32 | 2 | UCI |
| Breast Cancer 669 | Multivariate | 669 | yes | Integer | 10 | 2 | UCI |
| Credit Approval | Multivariate | 690 | yes | Mixed | 15 | 2 | UCI |
| Glass Identification | Multivariate | 214 | yes | Real | 10 | 7 | UCI |
| Mammographic mass | Multivariate | 961 | yes | Integer | 6 | 2 | UCI |
| IRIS | Multivariate | 150 | yes | Real | 4 | 3 | UCI |
| Wine Recognition | Multivariate | 178 | yes | Mixed | 13 | 3 | UCI |
| Human Activity Recognition | Multivariate, Time-Series | 10299 | yes | Real | 561 | 6 | UCI |
| Toy-data | Multivariate | 1500 | yes | Real | 2 | 5 | Self |
| **UCI- University of California Irvine's Machine Learning Repository; Self- Artificially generated dataset** | | | | | | | |

*Appendix A.5. Multi-Class ROC Analysis with RANs Modeling*

This study is carried out by two processes, first the input true-labels are transformed into a separate vector of binary labels, individually for all Abstract nodes (i.e., 1 for class c1, 0 for all other classes), second, calculating the confidence score for each instance of the input data (or test-data). Both processes are described as follows:

1. **Node-wise binary transformation of True-Labels**: For example, suppose there are three classes (c1, c2, c3) represented by three abstract nodes (n1, n2, and n3) in RANs model at Layer-1, and let true-label be [c1, c2, c2, c1, c2, c3, c3] for 7 test instances, then for node n1 label will be [1, 0, 0, 1, 0, 0, 0] where 1 represents class c1, and 0 depicts others (i.e., c2, and c3).

2. **Node-wise confidence-score calculation**: This is calculated by averaging activation-value and confidence-indicator of activation for an input instance at an Abstract node. Activation-value is an individual activation of an activation vector obtained by propagating up the data using UAP mechanism of RANs whereas, confidence-indicator is calculated by min-max normalization operation of activation vector. For example, after UAP operation each node (n1, n2, and n3) receives activation [0.89, 0.34, 0.11] (a vector of activation), and confidence-indicator is min-max ([0.89, 0.34, 0.11]) = [1.0, 0.29, 0.0]. and the confidence-score for nodes n1 = (0.89 + 1.0)/2.0 = 0.95, n2 = (0.34 + 0.29)/2.0 = 0.32, and n3 = (0.11 + 0.11)/2.0 = 0.05.

**Table A5.** Dataset specific configuration details of models.

| Data | Algo | Configurations | Data | Algo | Configurations |
|---|---|---|---|---|---|
| Toy-data | RBM + LR | Lr = 0.000001, iter = 500, comp = 20 max_iter = 30, C = 70 | UCIHAR | RBM + LR | Lr = 0.06, iter = 500, comp = 10 max_iter = 10, C = 1 |
| | K-NN | n_neighbors = 30 | | K-NN | n_neighbors = 15 |
| | LR | max_iter = 10, C = 1 | | LR | max_iter = 30, C = 1 |
| | MLP | Rs = 1, hls = 10, iter = 250 | | MLP | Rs = 1, hls = 10, iter = 400 |
| | RANs | CLS = 5, Desired_depth = 1 | | RANs | CLS = 2, Desired_depth = 1 |
| | SGD | alpha = 0.0001, n_iter = 5, epsilon = 0.25 | | SGD | alpha = 0.1, n_iter = 10, epsilon = 0.25 |
| Mice Protein | RBM + LR | Lr = 0.1, iter = 500, comp = 20 max_iter = 30, C = 30 | Breast Cancer 569 | RBM + LR | Lr = 0.006, iter = 100, comp = 10 max_iter = 30, C = 1 |
| | K-NN | n_neighbors = 15 | | K-NN | n_neighbors = 30 |
| | LR | max_iter = 4, C = 0.00001 | | LR | max_iter = 10, C = 0.001 |
| | MLP | Rs = 1, hls = 10, iter = 300 | | MLP | Rs = 1, hls = 10, iter = 200 |
| | RANs | CLS = 8, Desired_depth = 1 | | RANs | CLS = 2, Desired_depth = 1 |
| | SGD | alpha = 0.1, n_iter = 10, epsilon = 0.25 | | SGD | alpha = 0.0001, n_iter = 5, epsilon = 0.25 |

**Table A5.** *Cont.*

| Data | Algo | Configurations | Data | Algo | Configurations |
|---|---|---|---|---|---|
| Breast Cancer 669 | RBM + LR | Lr = 0.001, iter = 100, comp = 10 max_iter = 30, C = 1 | Credit Approval | RBM + LR | Lr = 0.006, iter = 100, comp = 10 max_iter = 30, C = 1 |
| | K-NN | n_neighbors = 10 | | K-NN | n_neighbors = 30 |
| | LR | max_iter = 10, C = 0.001 | | LR | max_iter = 10, C = 0.001 |
| | MLP | Rs = 1, hls = 10, iter = 200 | | MLP | Rs = 1, hls = 10, iter = 200 |
| | RANs | CLS = 2, Desired_depth = 1 | | RANs | CLS = 2, Desired_depth = 1 |
| | SGD | alpha = 0.0001, n_iter = 5, epsilon = 0.25 | | SGD | alpha = 0.0001, n_iter = 5, epsilon = 0.25 |
| Glass Identification | RBM + LR | Lr = 0.001, iter = 400, comp = 10 max_iter = 30, C = 5 | Mamographic Mass | RBM + LR | Lr = 0.01, iter = 500, comp = 20 max_iter = 30, C = 5 |
| | K-NN | n_neighbors = 15 | | K-NN | n_neighbors = 30 |
| | LR | max_iter = 5, C = 0.00001 | | LR | max_iter = 5, C = 1 |
| | MLP | Rs = 1, hls = 10, iter = 200 | | MLP | Rs = 1, hls = 10, iter = 250 |
| | RANs | CLS = 2, Desired_depth = 1 | | RANs | CLS = 2, Desired_depth = 1 |
| | SGD | alpha = 0.01, n_iter = 10, epsilon = 0.25 | | SGD | alpha = 0.0001, n_iter = 5, epsilon = 0.25 |
| IRIS | RBM + LR | Lr = 0.01, iter = 1000, comp = 20 max_iter = 30, C = 5 | Wine Recognition | RBM + LR | Lr = 0.01, iter = 500, comp = 20 max_iter = 30, C = 50 |
| | K-NN | n_neighbors = 15 | | K-NN | n_neighbors = 15 |
| | LR | max_iter = 10, C = 1 | | LR | max_iter = 10, C = 0.01 |
| | MLP | Rs = 1, hls = 10, iter = 400 | | MLP | Rs = 1, hls = 10, iter = 300 |
| | RANs | CLS = 3, Desired_depth = 1 | | RANs | CLS = 3, Desired_depth = 1 |
| | SGD | alpha = 0.01, n_iter = 10, epsilon = 0.25 | | SGD | alpha = 0.01, n_iter = 10, epsilon = 0.25 |

LR-Learning Rate; iter-Iterations; comp-Number of Hidden Components of RBM; RS-Random State; hls = Hidden Layer Sizes; CLS-Number of clusters at the input layer of RANs.

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
