# Peer review of "Exploring Geometric Feature Hyper-Space in Data to Learn Representations of Abstract Concepts"

_applsci, doi:10.3390/app10061994_

Round 1

Reviewer 1 Report

It is original to generalize and describe the abstract concept through the RAN algorithm proposed in this paper. In addition, the research, the proposed algorithms, and the mathematical techniques are well documented in the overall paper format. Therefore, I highly recommend this paper.

Author Response

Dear Reviewer,

We would like to take this opportunity to thank you for your time. We have carefully reviewed the comments and have revised the manuscript accordingly. We believe that the revised paper is a significant improvement over the manuscript. Specifically, we have:

(1)     We have rewritten the sections of the abstract.

(2)     We have revised some sentences with grammatical mistakes.

Our responses are given in a point-by-point manner to the comments of the reviewer. The responses are made to the comment in bold fonts in black color and pointers to the changes made in the manuscript are highlighted in the response with italic font and red color. The changes in the manuscript can also be seen through the blue color highlighting in content and bold highlighting for changes in tables. For the detailed response please see the attachment.

Kind regards,

Rahul Sharma.

(on behalf of all the authors)

Reviewer 2 Report

This paper presented a regulated activation network based learning model for abstract concept representation. The work was conducted based on previous research and the model was evaluated using classical machine learning data sets. The findings are consistent with the authors' hypothesis. Evaluation on more complicated data sets should be added. Also, more explanation about the "concept" term in psychology and neurobiology and how they are usually simulated using machine learning can be added so that the readers without related background can understand the contents quickly.

Minor errors exist in the writing: (1) in table 5, "mamographic" should be "mammographic", (2) in line 469, reference number is missing, (1) in line 194, "...method is used a concept identifier..." needs an "as" after "used" (4) in line 271, "shows" should be "show".

Author Response

Dear Reviewer,

We would like to take this opportunity to thank you for your time. We have carefully reviewed the comments and have revised the manuscript accordingly. We believe that the revised paper is a significant improvement over the manuscript. Specifically, we have:

(1)     We have rewritten the sections of the abstract.

(2)     We have revised some sentences with grammatical mistakes.

(3)     In the introduction, we supplemented the specific objectives and methods used in this paper. 

(4)  In section Modeling with RANs, we made significant changes to make the content more readable and modeling more understandable. We re-simulated the modeling with a Toy-data problem to explain the RAN’s methodology (as reflected in Section 3 “Abstract Concept Modeling with RAN´s”). The results presentations are improved by highlighting the best performing model and RAN’s results (change can be seen as Bold words in Table 6). Limitations and assumptions are also added to the article to provide more clarity to the readers (Section 3.1 “Assumptions and Boundaries”).

Our responses are given in a point-by-point manner to the comments of the reviewer. The responses are made to the comment in bold fonts in black color and pointers to the changes made in the manuscript are highlighted in the response with italic font and red color. The changes in the manuscript can also be seen through the blue color highlighting in content and bold highlighting for changes in tables 

Kind regards,

Rahul Sharma.

(on behalf of all the authors)

Reviewer 3 Report

The proposed method is a hybrid method that combines symbolic, distributed and spatial representations and use abstract concepts. The model uses a graph-based topology and not suitable for image datasets suitable. The proposed method has four basic steps. However, please see my comment below:

How to determine the number of groups for a dataset for K-Means in the concept identification process? You also mentioned any other clustering algorithm can be used. Did you use any other clustering algorithm that does not require any user inputs?

From Table 2, Table 4, and Table 5, the overall performance of existing techniques is better than RAN. Any idea why?

Please add a table that will show a summary (such as total records, total attributes, number of categorical attributes, number of numerical attributes, size of class attribute) of the datasets used in the experimentation.

Please provide complexity of the technique. If possible, please compare the complexity of RAN with the complexities of the existing techniques.

Author Response

Dear Reviewer,

We would like to take this opportunity to thank you for your time. We have carefully reviewed the comments and have revised the manuscript accordingly. We believe that the revised paper is a significant improvement over the manuscript. Specifically, we have:

(1)     We have rewritten the sections of the abstract.

(2)     We have revised some sentences with grammatical mistakes.

(3)     In the introduction, we supplemented the specific objectives and methods used in this paper. 

(4)     In section Modeling with RAN’s we made significant changes to make the content more readable and math more understandable.  We re-simulated the modeling with a Toy-data problem to explain the RAN’s methodology (as reflected in Section 3 “Abstract Concept Modeling with RAN´s”). The results presentations are improved by highlighting the best performing model and RAN’s results (change can be seen as Bold words in Table 6). Limitations and assumptions are also added to the article to provide more clarity to the readers (Section 3.1 “Assumptions and Boundaries”).

Our responses are given in a point-by-point manner to the comments of the reviewer. The responses are made to the comment in bold fonts in black color and pointers to the changes made in the manuscript are highlighted in the response with italic font and red color. The changes in the manuscript can also be seen through the blue color highlighting in content and bold highlighting for changes in tables 

Kind regards,

Rahul Sharma.

(on behalf of all the authors)

Round 2

Reviewer 3 Report

No comments